# The Evolution of Urban Spatial Structure in Brasília: Focusing on the Role of Urban Development Policies

**Cayo Costa and Sugie Lee *** 

Department of Urban Planning and Engineering, Hanyang University, 222 Wangsimni-ro, Seongdong-gu, Seoul 04763, Korea; cayo.costa@gmail.com
* Correspondence: sugielee@hanyang.ac.kr; Tel.: +82-2-2220-0417

**Abstract:** Many cities evolve over time, but some are designed from scratch. Brasília is presented as a unique case on urban planning for having been built from figuratively nothing, based on a design concept that was the brainchild of Brazilian urbanist Lucio Costa. The present study aimed to analyze the interrelation between urban planning and spatial structure change over time to understand the role of urban development policies on the spatial organization of Brasília. The study was conducted based on three interrelated aspects: (1) The intentions of the plans, (2) territorial governance, and (3) external conditions. The results showed that the circumstances of territory occupation—characterized by a polycentric development system with dispersed satellite cities economically dependent on Brasília—have been gradually replaced by strategic development policies, mainly influenced by social and political driving forces. Accordingly, this research suggests a reconsideration of the scale of development instrumentations based on a better understanding of the metropolitan area of Brasília as a unique structure by strengthening its interrelations and seeking better coordination of interests and adaptability of governance processes.

**Keywords:** urban development policy; urban form; spatial inequality; spatial structure; Brasília

## 1. Introduction

In the last century, certain countries have planned to transfer or even established new capital cities, following the steps of Washington and Canberra, or developing a town which already exists. More recently, Egypt and South Korea began developing their new capital cities. However, the decision to relocate a capital city is not simple, and the cost must be considered, as well as the fact that, as shown by Gottmann [1], capital cities act as articulators between different regions of the country. The rearrangement of these complex structures is complicated.

Therefore, it is imperative for planners and policymakers to understand the dynamics of the spatial structure of these new cities, and urban development policy due to the common arguments that planning influences spatial structure, particularly in urban areas. This transformation differs according to the speed and intensity of urbanization processes. In fact, the dynamics of transformation are a result of the relationship between several factors—political, cultural, natural, technological, and economic factors—and their agents [2,3]. Although a great part of the existing research has focused on natural and economic evidence to analyze spatial changes, recent studies have begun to pay special attention to the role of urban development policies as important drivers of spatial transformation patterns [4,5], highlighting the importance of developing better land use models to support urban planning.

Urban development tools are multidisciplinary and, therefore, have several objectives and scales, including land use plans, master plans or strategic plans. For various purposes, policymakers pursue conducting urban development with the aim of promoting the sustainable growth of regions [6] or in response to rapidly increasing housing demands [7]. Regarding spatial planning in Brazil, discussions

of systematic public policies toward the process of urbanization first occurred in the 1960s due to massive immigration and rapid urban growth. According to Veloso [8], the model for urban policies in this period was prescribed by the state due to political and financial support from the federal government. Before this period, the Brazilian territory was more simplistically considered a place of work, residence, and exchanges [9].

The work of Deák [10] shows that urban planning in Brazil was recognized as a set of actions around the spatial organization of urban activities, which were not to be settled or guided by the market. In other words, conception and application in urban planning were meant to be assumed by the state, and from this context, the city of Brasília emerged. The plan for a new capital was associated with the process of national integration through the implementation of the first National Road Plan (1951), promoting access and occupation of the hinterlands, which accelerated the emergence of small villages and the construction of new cities [11]. According to the numbers of the Brazilian Institute of Geography and Statistics (IBGE) National Census [12], the northern and west-central regions represented only 6.8% of the national population in 1960. In 2010, that percentage reached 15.9% of the national population.

Brasília is considered ground zero in the national road system, which explains the close relationship between the road network system and urbanism in the pilot plan of the new capital of Brazil [13] (pp. 230–239). As for the role of Brasília in the master plan for regional development, Costa [14] (p. 3), the author of Brasília's masterplan, highlighted in the project report that "Brasília would not be shaped in regional planning, but rather would be the cause of it." In other words, development of the Federal District would be defined by state intervention in the territory according to the guidelines proposed by Lucio Costa.

Although the evolution of Brasília's spatial structure is typically described as a product of the application of urban development processes over time, we cannot ignore other driving forces at work within the region. Friedmann [15] calls for non-Euclidean planning in a world of "many space-time geographies." He also argues that planning is the "real time" of everyday events, rather than general strategies. Following the same logic, the work of Graham and Healey [16] criticizes one-dimensional treatments, arguing that planning must consider relationships and processes rather than objects and form. Several researchers and experts consider space a social construction [17,18]. It is important to understand that the process of spatial production and organization is in constant transformation. Thus, uncertainty is intrinsic and must be considered as an important factor. Recent research shows that the combination of historical maps, geographic images, and population features can provide fundamental information on living space changes, as well as how changes stand to affect our future environment [19]. Studies also adopt document-analysis methods to evaluate the impact of planning and public policies in narratives of spatial structure transformation [20].

Therefore, the purpose of this paper is to examine the relationships between the transformation of spatial structure and urban development policies in Brasília, enabling the evaluation of potential strategies for future courses of urbanization. This study is organized into five sections. The opening introduction is followed by the sections of literature review and methodology. Section 4 examines the urban development policies in the Federal District, together with how those policies influenced the transformation of urban spatial patterns in Brasília over time and integrated analysis of this relationship. Finally, conclusions and discussion are addressed in the end.

## 2. Literature Review

The relocation of capital cities has come out in countries with distinctive economic development and been ruled by different political systems. According to Vale [21], more than three-quarters of the capital cities in 1900 were not serving as state capitals in 2000. Some purpose-built cities emerged on a tabula rasa, such as Brasília, Abuja (Nigeria), and Putrajaya (Malaysia). In contrast, cities like New Delhi (India) and Islamabad (Pakistan) have been developed adjacent to prior ones. Therefore, the features, actions, and ideas behind new capital cities require special attention.

The social and spatial structures of any given area play important roles in the evolution of new human settlements, emerging from both planning and spontaneous circumstances. These definitions are interrelated to the point where each system affects the other regarding characteristics and management. In land-use change science, planning is consistently identified as a political driver [22]. Current studies adopt the idea of driving forces as a framework for analyzing the causes, processes, and consequences of spatial changes. This approach has become an effective tool for evaluating urban development policies [23].

According to Bürgi et al. [2], five driving forces—which can be classified as political, cultural, natural, technological, and economic forces—determine an actor's decision making. In determining land-changes, for example, socioeconomic necessities are articulated in political programs and policies; thus, socioeconomic and political driving forces are strongly interconnected. Many analyses on political forces have been conducted through qualitative and quantitative evaluations of the impacts of urban policies on urban spatial changes [2,24,25]. Pagliarin [26] studied the relationships among suburban land-change patterns, political processes, and urban planning regulations, demonstrating how the dispersed metropolitan structure derives from local planning policies conducted by municipal governments via land-use micro transformations. Although several studies have explored the impact of urban policies and other actors on urban spatial changes, it remains difficult to conceptualize the role of spatial planning because of the complexity of the theme.

Due to government control of land within the Federal District and the lack of a regional plan, studies on spatial structure have focused on the pilot plan of Lucio Costa for the city [27,28]. In both previous analyses, Bertaud criticized the ideology that drives land use for having produced mild cases of population dispersion. In a different scenario, Moser [29] identified a spatial segregation issue based on racial identities in Malaysia's new capital city. According to this study, Putrajaya's design emphasizes the Muslim identity while excluding non-Muslims, which does not provide a great deal of flexibility for changing needs and demographic change. Other driving forces cannot be ignored; external and socioeconomic conditions influence the development and application of urban planning. The city of Brasília was developed to be a monocentric city, but due to great demographic growth in the early years, the urban area ended up expanding according to a model of polycentric occupation, with satellite cities scattered throughout the territory.

In a study on dormitory towns, Goldstein and Moses [30] highlighted the issue of commuting, especially the use of private cars and residential locations (usually distant from the city center), which increase transportation costs and lower housing prices with distance from the center. Ficher [13] (pp. 230–239) showed that automobile use was incorporated into the spatial plan of Brasília, thereby promoting a city molded by hierarchical and specialized traffic routes. In a comparative discourse analysis between two distinguished cities, Brisbane and Hong Kong, Leung et al. [31] explored the effects of the peak oil discourse in influencing urban transport policy and showed how transport policy is highly political. Consequently, roads and transportation, representing a key technological driving force in urban planning, play an important role in shaping metropolitan areas. This junction between the use of roads and dispersed satellite cities apparently shaped the unique spatial structure of Brasília.

According to Meijers et al. [32] (p. 18), "The establishment of a polycentric urban region as an actor has to deal with a large number of public and private actors, all having their own goals and preferences and often having differences in procedures, culture and power, perceived and real." However, Burger and Meijers [33] showed that most metropolitan areas present more morphologically polycentric than functionally polycentric patterns and that this difference is explained by size, external connectivity, and degree of self-sufficiency of the major center.

Regarding the local governance of capital cities, some urban centers perform national functions, while others perform more local functions [34]. Kaufmann [35] conducted a study comparing the locational policy agendas of Bern, Ottawa, The Hague, and Washington, D.C., revealing that local autonomy constraints, such as city budgets, are more common in purpose-built capitals than in purposely selected capitals. Consequently, secondary capital cities, such as Washington, D.C., Ottawa,

and Brasília, tend to request compensation payments, and elaborate development-oriented policies agendas. In addition, local governments are central actors in urban governance arrangements, since they lack an industrial history and strong private agents. The challenge of these cities is to find the equilibrium between government–market interests.

However, the methodology for evaluating long-term urban development remains necessary. Through GIS retrospective analysis, García-Ayllón [36] examined the evolution of the land market and the resulting urbanization in La Manga, a city created in 1963 out of nowhere as part of a strategy to develop tourism in Southeastern Spain. The analysis criticized the La Manga urban process as a tourism product market from the perspective of supply and demand for land, which ended up slowing the value of land, generating overcrowding, and aggravating the road traffic.

Following Adams [37], the relationships and interactions among agents such as developers, politicians, and landowners shape urban development processes. On the other side of the argument, Anas et al. [38] focused their studies on the relationship between urban spatial structure and market forces. The authors claimed that continuing decentralization represents a more polycentric form of urbanization, with subcenters that depended on an old central business district (CBD). According to Anas et al. [38] (p. 1), some subcenters are older towns, which were gradually incorporated into expanded urban areas. Others, by contrast, are distant from city centers, having been spawned at nodes of urban transportation networks, and are usually known as "edge cities."

To better understand these dynamics, several studies have evaluated and mapped spatial patterns and tendencies in metropolitan areas by analyzing land cover changes [39,40] and population patterns [41,42]. Ihlandfeldt [43] investigated the spatial distribution of jobs in Atlanta metropolitan areas, with the results indicating that people—regardless of their race or employment status—have poor access to jobs, a fact that is attributed to residential segregation. In reference to Brasília, Holanda et al. [44] discussed the main attributes of the metropolis concerning the economics of urban sprawl. Through several indices, the authors measured fragmentation, dispersion, and eccentricity within the region, showing how these features have negative consequences for socio-spatial segregation in Brasília. The study revealed a positive correlation between family income and distance from the CBD, and an arrangement wherein low-income families tend to live farther from the city center, which is a typical characteristic of Brazilian cities. In a recent study conducted by Pereira and Schwanen [45] on commuting time in Brazil among metropolitan regions, it was affirmed that in the Federal District, journey-to-work trips are 75% longer between the poorest population decile and the richest decile. In another study conducted in Beijing, Lin et al. [46] suggested that the development of economic and employment clusters could influence employees' commuting times.

Although there are many studies on the characteristics of urban spatial structures in Brasília, the outcomes of urban development policies are rarely evaluated. Instead, measures of given impacts usually focus on user satisfaction [47]. Planning evaluation is an important stage of the spatial planning process and is fundamental to understanding the role of urban policies. Gordon [48] analyzed the 1915 Report of a General Plan for Canada's capital and discussed the transition in planning practices from "city beautiful" to "city scientific," showing that the aesthetic approach ignored significant aspects, such as housing and social issues. Nevertheless, few empirical studies have focused on urban planning implementation, including which of its aspects are important to measure. Lai and Baker [49] claimed to develop a theory of planning process and the need for strategic regional plans and strategic planning bodies in growing economies. Moreover, as pointed out in the work of Kinzer [50], disagreement on a clear definition of what is considered successful planning implementation is a barrier to reaching a better understanding of the role of public policies. Because the relationship between the evolution of spatial structure in Brasília and urban development policy is not addressed in previous articles, this study aims to provide a better linkage between these two topics.

## 3. Methodology

### 3.1. Study Case Area

Many cities evolve over time, but some are designed from scratch. In 1915, the urban planner Edward Bennett launched the "city beautiful" plan for Canada´s new capital, Ottawa. The plan included the cities of Ottawa and Hull. When Queen Victoria designated Ottawa as Canada's capital in 1857, Ottawa was a small lumber town with the population of 10,000–12,000 [48].

Another case is Pakistan's new capital city. From 1959 to 1963, Pakistan was also conceiving the master plan of a new capital city as a part of a large metropolitan area by integrating it to the city of Rawalpindi as a twin city [51]. However, the plan was never put into practice due to the lack of institutional development. Recently, aiming at balanced national growth between the Seoul metropolitan area and local regions, South Korea developed a new administrative city of Sejong, which is 145km away from the city center of Seoul. Sejong is based on the concept of sustainable development with transit-oriented development (TOD) and traditional neighborhood development (TND). The city's expected population is 500,000 [52]. Brasília is presented as a unique case on urban planning for having been built from nothing in a depopulated area at the end of the 1950s.

The initial layout of Brasília was the brainchild of Brazilian urbanist Lucio Costa. Moreover, the city is the center of the former political and administrative power of Brazil, a region known as the Federal District. Facing geopolitical and economic changes, the region was declared a World Heritage Site by the United Nations Educational, Scientific, and Cultural Organization (UNESCO) in 1987 and has undergone various planning approaches. With its greatly expanded urban area, the city today is much more complex than the city represented in the pilot plan of Lucio Costa (see Figure 1).

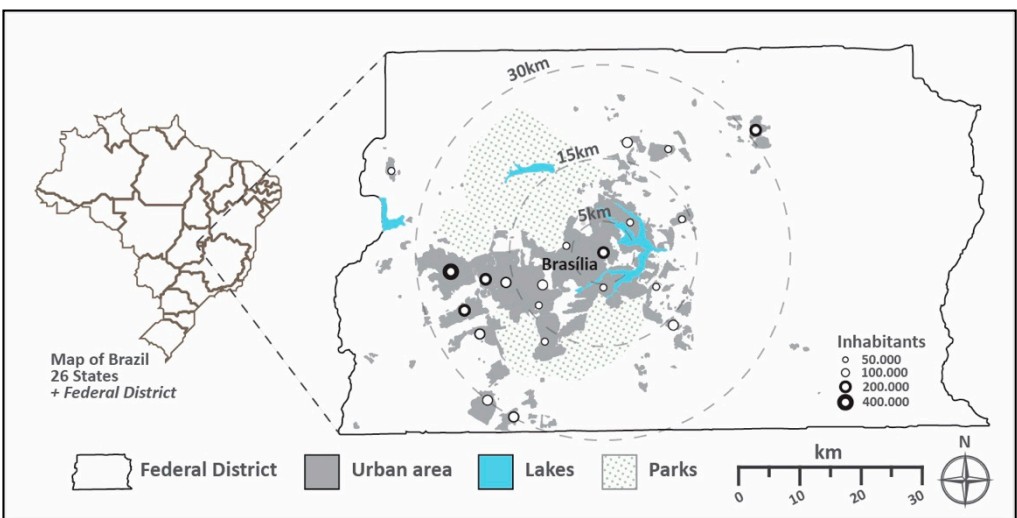

**Figure 1.** Map of the Federal District. **Source:** Data collected from 2015 District Household Sample Survey (Pesquisa Distrital por Amostra de Domicílios: PDAD); Urban cover area from Secretary of State for Territorial and Housing Management (Secretaria de Estado de Gestao do Territorio e Habitacao: SEGETH).

The new capital of Brazil was intended to be a monocentric city. Instead, the urban area ended up following a model of polycentric occupation via the implementation of satellite cities scattered throughout the territory. The model proposed by Lucio Costa represented only a small part of the urban picture in today's Federal District; the regions officially called "Brasília," "South Lake," and "North Lake" accommodate only 12% of the metropolitan population [46]. The area of influence of Brasília expands to cities of the neighboring state, near the limits of Brasília. Its status as the third largest Brazilian metropolis with an estimated population of 3,039,444 in 2017 (IBGE) was achieved due to urban development strategies employed in the Federal District. Consequently, this study shows

how urban development processes are fundamental criteria of the connection between urban growth and spatial organization in the region.

Moreover, until 1960, the Brazilian modernist movement was well respected on an international scale. Between the decades of 1930–1960, Brazil was known as the world capital of modernism, with the claim that "nowhere else was modernist architecture so enthusiastically adopted as a national style" as in Brazil [53] (p. 2). In 1929 and 1936, Le Corbusier visited Brazil to work on a project in Rio de Janeiro, during which he promoted some conferences that helped to advance his ideas in the country and powerfully informed his own practice [54] (pp. 113–115). The Swiss architect had an enormous influence on the work of Lucio Costa, Oscar Niemeyer, and other architects of that generation.

### 3.2. Data and Methodology

Government policies can be considered as the main driving forces of growth in urban areas [55]. Nevertheless, to quantify governance and conceptualize the role of urban development policies on the spatial structure is a great challenge [56], partly due to limited knowledge. In addition, the challenge is due to uncertainty about the definition of the production of space, as Hillier [57] (p. 30) stated that "unexpected elements come into play and things do not work out as expected in strategic planning practice." In this way, space is a social construction. On the other hand, Briassoulis [58] considers policy-making and planning as technical, stylized, and top-down activities.

Concerning analysis of the role of spatial planning on land change, Hersperger et al. [59] proposed a framework based on three important interrelated elements (see Figure 2). Namely, (1) the intentions indicated in planning maps or text, together with the built environment as envisioned, (2) territorial governance (in other words, the processes by which policies that involve the coordination of different actors and interests are developed), and (3) any external conditions that might affect the development and/or implementation of a given plan (for example, unstable economic or political situations). These conditions can reinforce path-dependent policies or result in the selection of new paths.

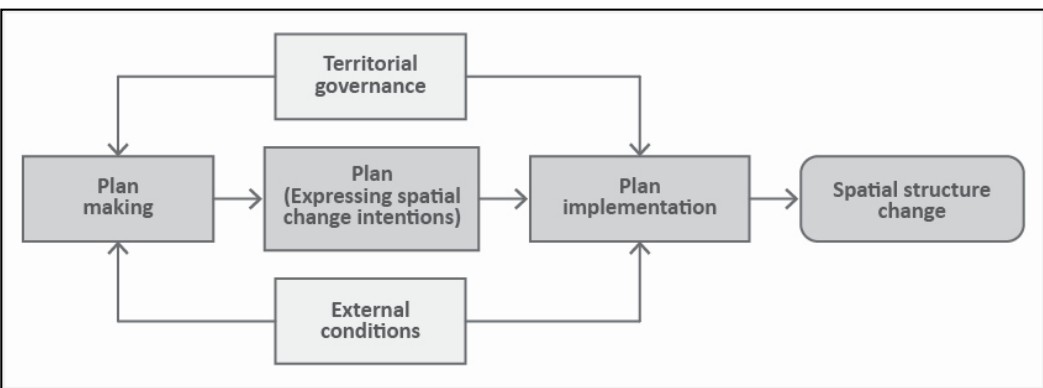

**Figure 2.** Diagram of the impact of spatial planning on spatial structure change. **Source**: Diagram is adapted from Hersperger et al. (2018).

Regarding data sources, government policies and political factors were considered as indicators for analysis (see Table 1). In addition, the Territorial and Urban Information System (TUIS) developed maps on the urban expansion of Brasília from 1958 to 2015, from which important information was extracted about urban development changes and fragmentation over time. These maps are suitable for general urban analysis, but they cannot quantify levels of urbanization because the maps do not provide information on population and housing density [60]. This analysis is contextually rooted in the temporal evolution of the urban characteristics observed in Brasília, useful in describing the dispersed concentration of the population in specific polycentric patterns. The evolution of Brasília's spatial structure provided an ideal context to examine how space responds to political uncertainty and changes, and when development agendas converge with the public interests.

**Table 1.** Main urban policies implemented in the Federal District.

| Year | Urban Policy | Key Contents (Intentions) |
|------|------|------|
| **1956** | Urbanization Company of New Capital of Brazil | Outline planning and construction of Brasília, control of land use, and is responsible for the urbanization and infrastructure. |
| **1957** | Pilot Plan "Report" | The master plan of Brasília (details in Figure 3). |
| **1964** | National Bank of Habitation | It promotes national housing policies and the expansion of employment in construction sectors. |
| **1965** | Housing of Social Interest Society | Plan for the provision of social interest housing. |
| **1975** | Special Program for the Geoeconomic Region of Brasília | Seeks to expand the geoeconomic influence of Brasília and to develop the satellite cities. |
| **1978** | Territorial Organization Structural Plan | It considers the southwest axis as the main vector of urban growth. |
| **1985** | Territorial Occupational Plan | Promotes for the first time public participation and sets limits in all zones for urban expansion. |
| **1987** | "Brasília Revisited" Plan | Requires the preservation of main aspects of the original design and plans expansion through the creation of new blocks. |
| **1990** | Plan of Land Use | Defines and distinguishes between uses and activities in urban and rural zones. |
| **1992** | Land Use Planning | Promotes the occupation of urban voids between Brasília and satellite cities. |
| **1997** | Land Use Planning | Delimits environmental monitoring zones and promotes the creation of southwest industrial poles. |
| **2009** | Land Use Planning | The most recent plan creates an urban containment zone to control irregular growth. |

**Source:** Public Archives of the Federal District, Brasília.

Consequently, this research adopted the framework of Hersperger et al. [59] as a primary methodology. In other words, this study was mostly conducted through descriptive analyses including demographic and socioeconomic analysis, and spatial analysis of the evolution of urban growth in the metropolitan Brasilia area. In addition, this study analyzed the spatial characteristics of satellite cities as well as Brasília over time. Finally, this study describes the relationships between urban development policies and the evolution of urban spatial structures of the metropolitan Brasília area.

Population data corresponding to each period was derived from National Census reports (1960, 1970, 1983, 1991, 2000, and 2010) and Household Sample Surveys (2013 and 2015). Other data were borrowed from existing studies, documents, and related figures. Analysis of spatial evolution and population growth was developed by contrasting the numbers for corresponding years in order to obtain a long-term growth rate. This way, this research sought to focus on critical analysis of existing literature, tracing the transformation of urban forms connecting public policies with their consequences for the development of the region.

## 4. Analysis

### 4.1. Origination of Brasília

Brasília is not a conventional city. Brasília did not originate spontaneously from any previous occupation of space as a result of economic, social, and political processes inherent in urban dynamics. Instead, it is a city that emerged from an idea transformed into a design. The city was built as a symbol of modernity under the precepts of modernist ideals. The physical shape and structure of the metropolitan area of Brasília have been gradually modified according to zoning regulations, mainly via housing development policies. What is more, Brazilian society faced trends of internal migration on a

scale never observed before, especially between the decades of 1950 and 1970. At this point, Brazil had officially become an urban nation due to the intensification of the industrialization process initiated mainly by the opening of the economy to foreign capital [11].

As its first act in the implementation of a new capital, the federal government required the transfer of the capital from Rio de Janeiro to the new city, together with the creation of the Urbanization Company of the New Capital of Brazil (NOVACAP). The company established to control land use and directly execute or contract companies for projects on behalf of the state. In addition, NOVACAP was put in charge of a national contest for the elaboration of the master plan of Brasília. The federal government envisioned a plan for 500,000 inhabitants maximum, along with a roadway and railway connecting Brasília to Anápolis (taking into consideration the pre-existing location of the airport) in the southwest area of the Federal District.

Urban planner Lucio Costa, the engineer of the winning project, developed a plan that masterfully included all the central elements of the territory. The original project was inspired by concepts of urban modernism, idealized from a rational and functional plan based on a transport system of roadways [61]. Accordingly, the Pilot Plan "Report" on Brasília (1957) is considered the first application of urban regulation in the region. In addition to the creation of the satellite cities—later termed "administrative regions"—these first elements, as envisioned by Costa, serve to confirm the role of Brasília as a planned center, which was intended to influence the production of space within the future metropolitan area.

Regarding the Pilot Plan's spatial structure, Lucio Costa divided the built space into four sectors: Monumental, residential, social, and bucolic (see Figure 3). The architect did not prohibit the mixed use of sectors; however, the distribution applied both in Brasília and the satellite cities led to the sectorization of functions all over the territory. The theory of functionalism, upheld by Le Corbusier and arguably his greatest influence, established that in a "Contemporary City," everything is classified by function, with discrete functions occupying and characterizing separate sectors [7]. Moreover, in Brasília, the land was controlled by the state and administratively distributed, rather than sold on a free market.

Lucio Costa suggested that the NOVACAP urbanization company would act as a real estate developer and that the price index should follow the demand. In his own words:

> "I understand that the blocks should not be landed [and], rather than the sale of lots, the state should provide land quotas, whose value would depend on the location, in order not to impede the current planning and possible future remodeling in the internal delineation of the blocks." [14] (p. 15)

Additionally, the author suggested an upfront evaluation of all proposed private projects in two stages—namely, "draft" and "definitive" projects, in order to promote better quality control of the built environment.

Complementing its applied functionalism, one of the foundations of the new city was its extensive road network, implemented over the whole territory as part of the Highway National Plan, of which Brasília was the center [11]. Carpintero [62] states that the road network of Brasília was based on specific functions established in the Charter of Athens, with the two structuring axes (the Monumental Axis and the South-North Axis) converging to the central area. Le Corbusier used the same concept in the Contemporary City, described as "Two great superhighways (one running east-west, the other north-south) form the central axes, intersecting at the center of the city" [6]. Accordingly, the work of Costa [14] (p. 15) states that "Because the framework is so clearly defined, it would be easy to build." The project prioritized the use of private automobiles as the main means of urban transportation, with highways at the core of the city.

Moreover, the project followed the guidelines of the garden city movement, characterized by a great proportion of green and open spaces and low-density occupation, thus lending the urban environment a park feeling [63]. The empty spaces in Brasília were considered elements of the modernist structure, and Lucio Costa justified them by saying that he was inspired by the immense lawns of English landscapes [62] (pp. 132–133).

The description above shows the foundation of the current spatial structure of Brasília, which the process of urban evolution divides into two distinguished periods—namely, the first two decades (the 1960s and 1970s), when the main satellite cities and road structure were implemented, and the second period (from 1980 to the present day), which has seen the consolidation of the metropolitan area.

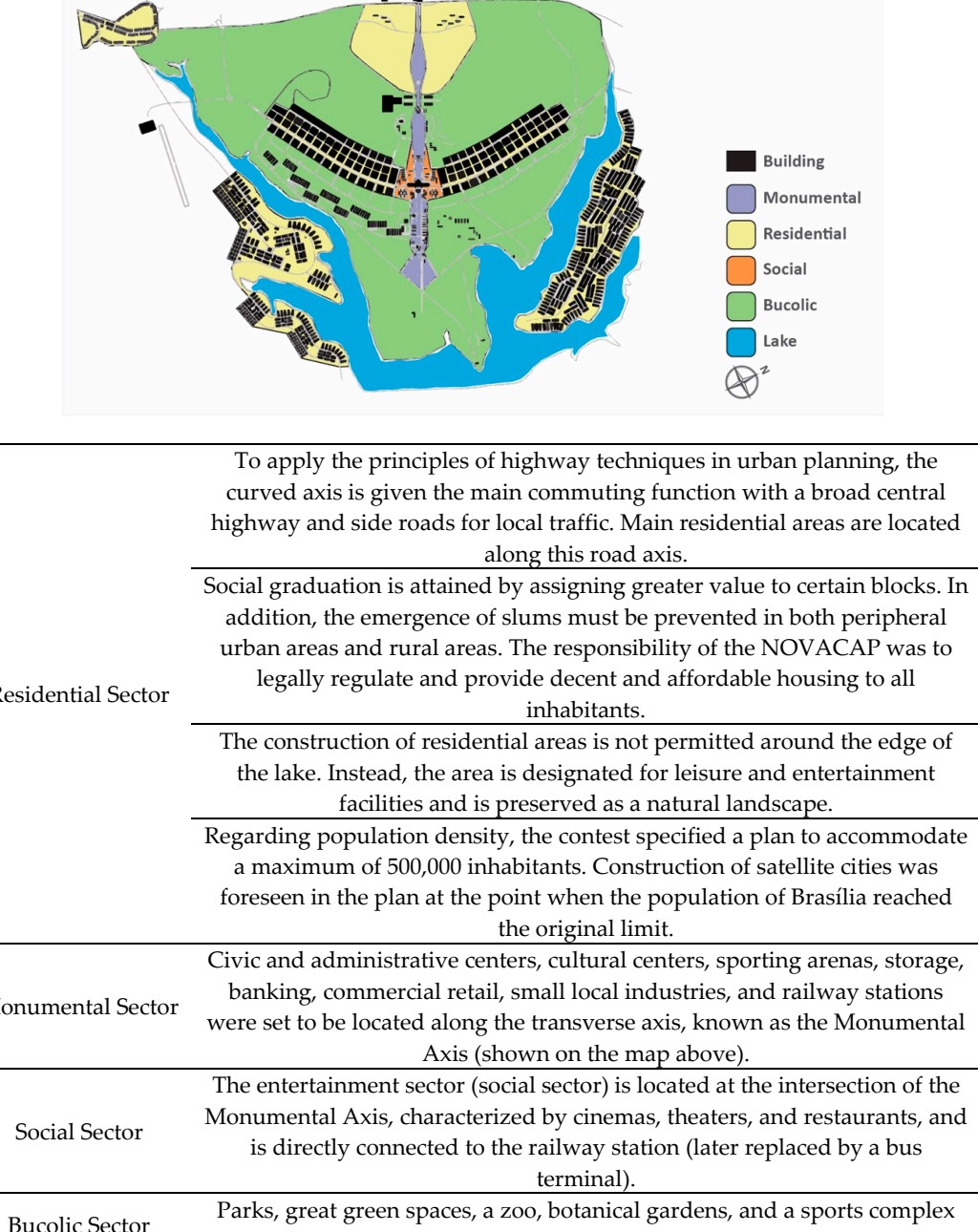

| | To apply the principles of highway techniques in urban planning, the curved axis is given the main commuting function with a broad central highway and side roads for local traffic. Main residential areas are located along this road axis. |
|---|---|
| Residential Sector | Social graduation is attained by assigning greater value to certain blocks. In addition, the emergence of slums must be prevented in both peripheral urban areas and rural areas. The responsibility of the NOVACAP was to legally regulate and provide decent and affordable housing to all inhabitants. |
| | The construction of residential areas is not permitted around the edge of the lake. Instead, the area is designated for leisure and entertainment facilities and is preserved as a natural landscape. |
| | Regarding population density, the contest specified a plan to accommodate a maximum of 500,000 inhabitants. Construction of satellite cities was foreseen in the plan at the point when the population of Brasília reached the original limit. |
| Monumental Sector | Civic and administrative centers, cultural centers, sporting arenas, storage, banking, commercial retail, small local industries, and railway stations were set to be located along the transverse axis, known as the Monumental Axis (shown on the map above). |
| Social Sector | The entertainment sector (social sector) is located at the intersection of the Monumental Axis, characterized by cinemas, theaters, and restaurants, and is directly connected to the railway station (later replaced by a bus terminal). |
| Bucolic Sector | Parks, great green spaces, a zoo, botanical gardens, and a sports complex comprise the bucolic sector. |

**Figure 3.** Functionalism in Brasília described by the urban planner Lucio Costa, the author of the master plan of Brasília in 1957. **Source**: Costa (1957) [14].

### 4.2. Modernist and Centralized Period

Analysis of the urban policy implementation process is fundamental to understand the interrelation between urban growth and centralized political control in the Federal District. The pilot

plan predicted the long-term creation of orderly planned satellite cities in case the center city achieved its limit of 500,000 inhabitants over time [64]. Instead, construction of the satellite cities was precipitated by rapid population growth within the first years of the new city (see Table 2). The immigrants were in great part escaping from the historic drought that occurred in Northeastern Brazil between 1957 and 1958, which contributed to the proliferation of illegal settlements around the territory [10]. In 1960, the inaugural year of the new capital, the first three satellite cities of Taguatinga, Sobradinho, and Gama were already established with the purpose of accommodating both workers and residents of illegal settlements.

**Table 2.** Population and annual growth rate of Brasília and Brazil from 1960 to 2018.

| Years | Brazil | | Brasília | |
|---|---|---|---|---|
| | Population | Annual Growth Rate (%) | Population | Annual Growth Rate (%) |
| 1960 | 70,070,457 | 3.06 | 139,796 | - |
| 1970 | 93,139,037 | 2.91 | 537,492 | 14.42 |
| 1980 | 119,011,052 | 2.50 | 1,165,184 | 8.04 |
| 1991 | 146,825,475 | 1.94 | 1,601,094 | 2.90 |
| 2000 | 169,799,170 | 1.64 | 2,051,146 | 2.51 |
| 2010 | 196,834,086 | 1.27 | 2,570,160 | 2.28 |
| 2018 | 208,494,900 | 0.72 | 2,974,703 | 1.47 |

**Source:** IBGE National Census 1960–2018.

Taguatinga was strategically located near the department of the National Institute of Immigration and Settlement (30 km from the CBD) in charge of connecting workers with job opportunities [65] (pp. 85–94). Sobradinho (20 km from the CBD), also a destination for inhabitants of illegal settlements, soon became a common residence for federal workers. The city of Gama (38 km from the CBD) accommodated residents from illegal settlements and was built around the construction sites of Brasília. The design of Gama was inspired by the project that took third place in the national contest for the new capital. Bertaud [27] states that cities, as they grow in size, tend to change from a monocentric structure to a polycentric structure gradually. The work of Medeiros and Campos [66], however, shows how Brasília was born as a polycentric town.

The first satellite cities consisted of urban centers promoted by the state, designed as dormitory cities with most of the economically active population working outside the municipality. This policy continued for the next years, physically isolating low-income residents (see Table 3). Consequently, these dwellers were forced to face long-distance commuting, costly public transportation, and reduced access to urban scale economies [67].

**Table 3.** Monthly income range calculated based on 1980 monthly minimum wage rate.

| Cities | Monthly Income (Minimum Wage in Cruzeiro Real) | | | | | |
|---|---|---|---|---|---|---|
| | Total (Families) | No Wage (%) | $\frac{1}{4}$ to 2 * (%)(%) | 2 to 5 (%) | 5 to 10 (%) | 10 more (%) |
| Brasília | 97,042 | 1% | 9% | 17% | 21% | 50% |
| Taguatinga | 109,586 | 1% | 28% | 45% | 18% | 7% |
| Gama | 30,451 | 2% | 33% | 46% | 16% | 4% |
| Sobradinho | 15,714 | 1% | 44% | 38% | 22% | 12% |

**Source:** IBGE National Census 1980. * 1 unit of minimum wage = Cr$4149.6.

Brasília was initially planned as a car-oriented city with modernist design concepts. Regarding the road system, the government launched the Federal District Highway Plan in 1960 with the aim of integration, circulation, and distribution of local productions. Inspired by the American park-way system, the plan included 13 parkways linking regional and federal highways. Among 13 new parkways, one parkway called the Contorno Park Road (140km) around the city center was built as a physical barrier to control urban growth. This parkway was intended, in part, to preserve the pilot plan. Moreover, in 1961, a Conservation Unit, located in the National Park, was created from the need to protect the rivers supplying water to the federal capital and to maintain the natural vegetation. Comprising 30,000 hectares at the time, this unit also contributed to controlling urban growth in the north of Brasília.

The patterns of expansion in Brasília and the satellite cities ended up defining the model of dispersed urban growth for the next 20 years. Ferreira [68] argued that by not following the specifications of the original plan (which accounted for peripheral growth at a later phase due to natural expansion), Brasília optimized an organizational strategy of space.

In 1964, the country underwent tremendous political change at the hands of a military coup (the first major external condition affecting territorial governance). The period brought resurging interest in consolidating Brasília as the capital of Brazil following strong resistance from political opponents of Jucelino Kubitscheck, the ex-President responsible for the construction of Brasília [11]. Moreover, between 1960 and 1964, the city had seven different mayors, with public institutions facing several administrative and structural changes. During this period of political uncertainty, immigration was intense but soon was controlled by the military. As part of the strategy to restore investments made in Brasília, the Housing Finance System (law no. 4380, 21 August 1964) was created to attend to the national demand for housing, particularly in middle and low-income segments of the population. The National Housing Bank (NHB), as the central instrument of the Housing Finance System, had economic mechanisms for stimulating the acquirement and construction of social interest housing through private initiatives. The hallmark of NHB was communication between public and private sectors, which would oversee the production, distribution, and control of new dwellings to serve the greater need.

Also in 1964, as a strategy to consolidate the polycentric urban model, law 4.545/64 was created to divide the territory of the Federal District into administrative regions, including Brasília, Gama, Taguatinga, Brazlândia, Sobradinho, Planaltina, Paranoá, and Núcleo Bandeirante. In 1967, Guará was created. In 1969, Ceilândia was developed to accommodate 40,000 residents from an illegal settlement known as Vila IAPI (see Table 4). In contrast to previous cities, these two new towns marked the inauguration of a new urban strategy insofar as both were located within consolidated regions [10]. This new approach would ultimately be confirmed by the Territorial Organization Structural Plan of 1978. Regarding rising demands for high-income housing, the government expanded the regions of Lago Sul and Lago Norte as well as oversaw the creation of "Park Way" (all located within the limits of the road-park beltway) near the lake in southwestern Brasília.

**Table 4.** The first generation of satellite cities in Federal District (2015).

|  | Total | Taguatinga | Sobradinho | Gama | Guará | Ceilândia |
|---|---|---|---|---|---|---|
| Distance to CBD | - | 30 km | 20 km | 38 km | 12 km | 24 km |
| Area (ha) * | 12,856 | 2661 | 1504 | 2631 | 2367 | 3693 |
| Population (thousand) | 1015 | 207 | 62 | 134 | 133 | 479 |
| Density (pop. per ha) | 79 | 77 | 41 | 51 | 56 | 129 |
| Houses (thousand) (%) * | 314 (24.8%) | 69 (29%) | 19 (23%) | 41 (18%) | 46 (54%) | 139 (4%) |
| Develop. period | - | 1957 | 1959 | 1960 | 1967 | 1969 |
| Land developer | - | NOVACAP | NOVACAP | NOVACAP | NOVACAP | NOVACAP |

**Source:** PDAD 2015; * SEGETH 2015.

In the same period, in order to stimulate the investments in rural areas, several roads were built or expanded to access the production centers. One year later, in 1968, the first Transportation Master Plan of the Federal District was required by the Ministry of Transports and Secretary of Planning and was created. This plan was established following the Territorial Organization Structural Plan, excluding the city of Brazlândia, which was located far away from the city center and had a small volume of commuting at that time. However, due to the lack of renewal of the proposed guidelines, the first transportation master plan became obsolete.

Between the decades of the 1970s and 1980s, Brasília faced expansion around the satellite cities due to the construction of great residential neighborhoods through the Housing Finance System. The work of Ferreira [68] points out that the territory's periphery had accommodated 91% of all the low-income families in the entire region, which consisted of 570,000 inhabitants in 1973. Moreover, it is important to highlight that housing in Brasília was primarily reserved for public sector employees of the old capital, Rio de Janeiro [69] (pp. 64–65). The region of Cruzeiro was reserved for the military, and the "wings" were reserved for public service workers [70].

### 4.3. Strategic and Decentralized Period

In the early 1980s, the state developed two new satellite cities following a gap of 12 years from the establishment of the final city in the preceding wave (Ceilândia in 1969). The second generation of satellite cities started with Samambaia (1981), and engineer Riacho Fundo (1983) followed the economic block of Taguatinga and Ceilândia bordering the Taguatinga park road. In addition, an industrial sector was installed within Ceilândia's limits to attend the demand of jobs in the region, followed by the construction of an expressway directly connected to the Pilot Plan. In the following years, in order to meet the demands of the housing shortage, the government of Brasília implemented the administrative municipality of Samambaia, and later of Santa Maria, Recanto das Emas, São Sebastião, Paranoá, Riacho Fundo 1 and 2, and Candangolândia. In addition, the government oversaw the expansion of the pre-existing satellite cities.

In 1985, during the redemocratization of Brazil (following 20 years of military regime), the local government elaborated upon the Plan of Territorial Occupation. At this point, the new government proposed micro zoning and established the use of land according to two basic categories: Urban and rural soils. These new measures were a strict response to previous indiscriminate use of rural areas for urban purposes. In addition, two important features of this phase were the push for the formation of an urban agglomeration from Taguatinga and Ceilândia to Gama (see Figure 4), and criticism of the theory and practice of functionalism. Ten years after the first plan, the Public Transportation System of the Federal District was created to accommodate the demand of society in 1987. In 1991, the government took the first step towards a mass transport system, which was inaugurated in 2001, connecting the satellite cities of Samambaia, Taguatinga, Águas Claras, and Guará to the Pilot Plan. In 2006, Ceilândia opened its first station and was integrated into the existing transportation system.

In 1988, the new Federal Constitution (the second major external condition affecting territorial governance) granted political–administrative autonomy to the states, cities, and the Federal District. In addition, the constitution forced cities to formulate their own Urban Development Master Plans [71]. The Plan of Land Use, implemented in 1990, consolidated the definition of urban and rural zones, opening for the first time the possibility for private interests to participate in the parceling of land. The period also marks the return on investment in roads within the eastern sector, as well as maintenance of the road–park beltways. The work of Lopes [72] argues that this period was characterized by the dominance of initiatives of metropolitan actions, in which, time and again, will was shown to surpass rationality.

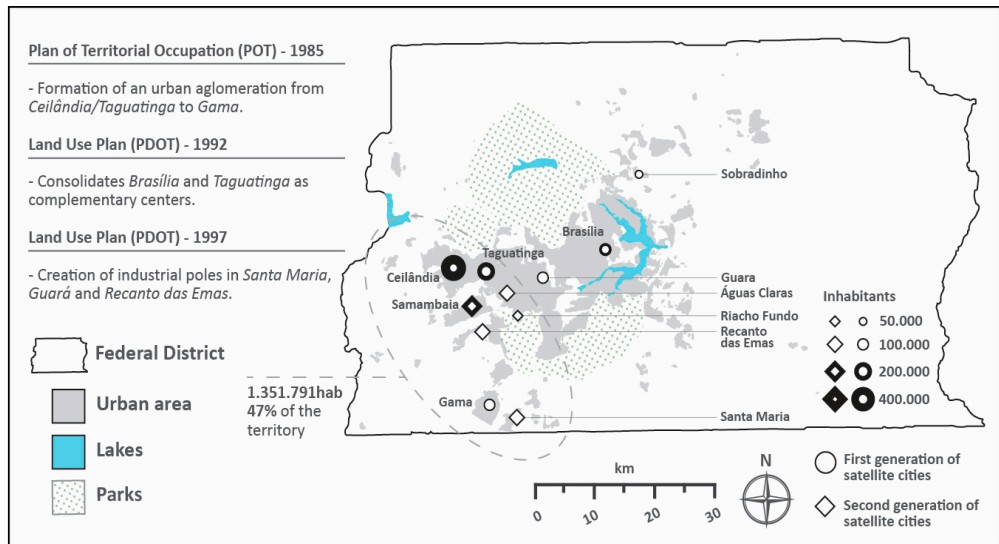

**Figure 4.** Urban agglomeration and economic decentralization promoted by urban development policies between the late 1980s and 1990s. **Source:** Drawn by authors, POT-1985, PDOT-1992/1997.

Regarding the impact of territorial governance on urbanization, another aspect of the new constitution (1988) was decentralization of urban management throughout the whole district toward the aim of greater independence in its localities. From this moment, each municipality was expected to form discrete departments of planning, which were to be in charge of developing zoning codes. In addition, the plan required the creation of jobs all over the district to improve the job-housing balance.

In the 1990s, the administrative regions of Sudoeste (in the central area of Brasília) and Águas Claras (between Guará and Taguatinga) were created to appeal to middle-upper and middle-income inhabitants. Águas Claras, in particular, is characterized by the verticalization of its residential buildings—the limits of which were modified in relation to Lucio Costa's original plan—thereby intensifying the density in this region [70] (see Table 5). Regarding the dispersed expansion (represented by the distant satellite cities), one of the objectives was to preserve the function of the federal government [73]. In 1992, the first Land Use Plan consolidated the region of the pilot plan and Taguatinga as complementary centers, connected by a system of mass transport. Along these lines, the plan also reinforced the satellite cities of Samambaia, Recanto das Emas, Gama, and Santa Maria as poles of secondary development.

**Table 5.** The second generation of satellite cities in the Federal District (2015).

|  | Total | Samambaia | Riacho Fundo | Santa Maria | Águas Claras | Recanto das Emas |
|---|---|---|---|---|---|---|
| Distance to CBD | - | 25 km | 18 km | 26 km | 19 km | 26 km |
| Area (ha) * | 8255 | 2468 | 466 | 2180 | 1895 | 1246 |
| Population (thousand) | 707 | 258 | 40 | 125 | 138 | 146 |
| Density (pop. Per ha) | 85 | 104 | 85 | 57 | 73 | 117 |
| Houses (thousand) (%) * | 207 (24.8%) | 69 (10%) | 13 (32%) | 34 (5%) | 49 (75%) | 42 (2%) |
| Development period | - | 1981 | 1983 | 1990 | 1992 | 1993 |
| Land developer | - | NOVACAP | NOVACAP | NOVACAP | NOVACAP | NOVACAP |

**Source:** PDAD 2015; * SEGETH 2015.

The strategy of satellite cities (cities developed beyond the limits of the road–park beltway) transformed the Federal District to a more dispersed spatial structure (see Figure 5). Concerning urban expansion along the northeast and southeast vectors, revision of the land use plan (1997) required rigid control throughout the region, after which housing districts were implemented, and urbanization was expanded through this vector. Additionally, the plans confirmed the west/southwest axis as a priority in the evolution of the new spatial structure. Regarding economic decentralization, the plan promoted the creation of an industrial pole in Santa Maria, a fashion pole in Guará, and a wholesale pole in Recanto das Emas (cities located in the southwest). Santa Maria and Recanto das Emas are both situated on the growth vector established in the Plan of Land Use from 1990.

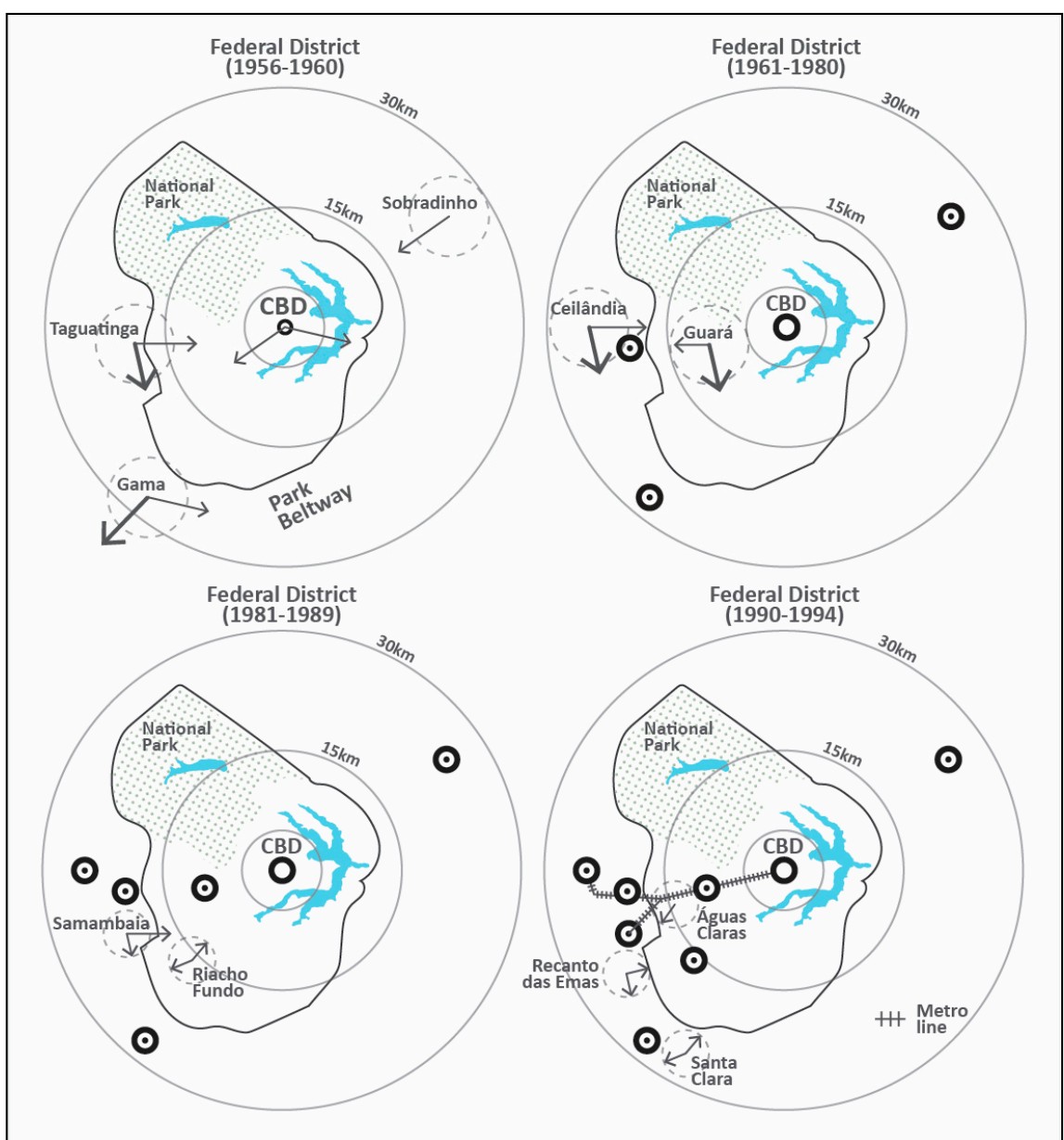

**Figure 5.** Evolution of the urban spatial structure (metro line works started in 1992 and completed in 2008). **Source:** Drawn by authors, SEGETH 1956-1994.

In 2009, the government launched a review of the land use plan (1997), confirming Brasília as a developing regional center and national metropolis. Regarding the novel policies of decentralization, the document reinforced the push for the creation of new projects in consolidated areas. Although past

land use plans had fought against dispersed growth, during the decades of the 1990s and early 2000s, the urban area growth rate was higher than population growth (see Figure 6, Figure 7, and Table 6).

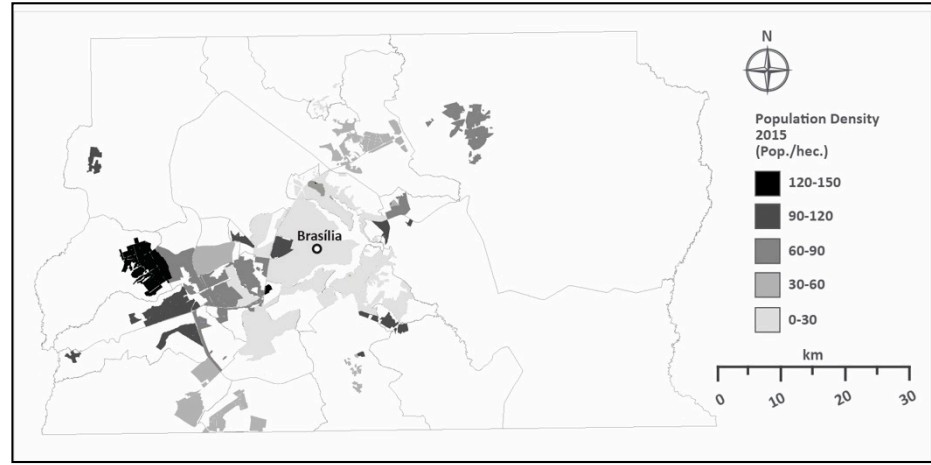

**Figure 6.** Population density of the Federal District (2015). **Source:** SEGETH (2015).

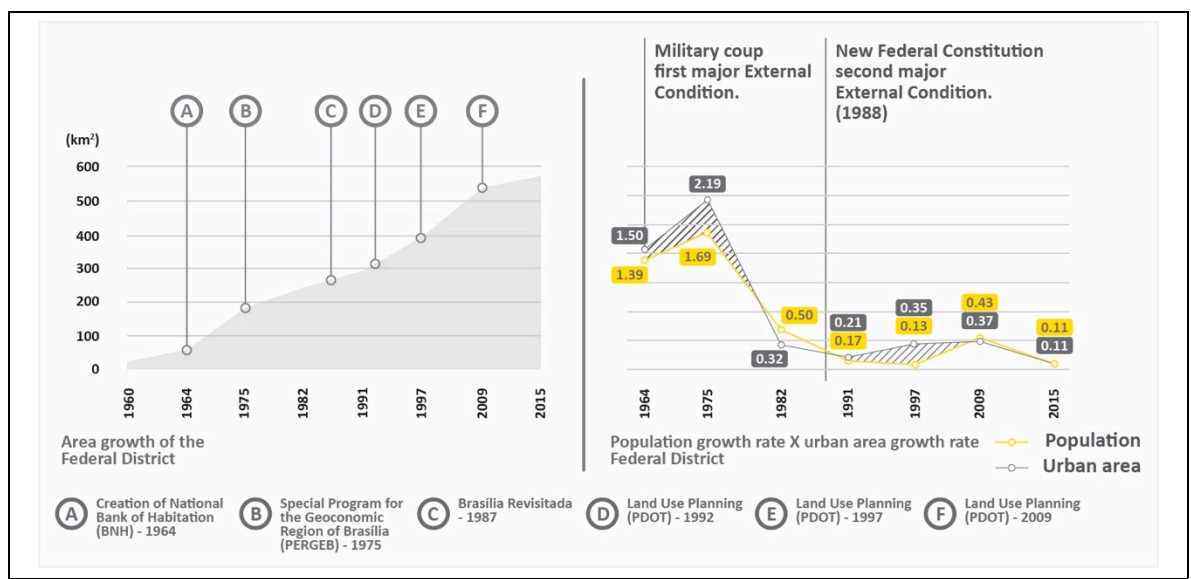

**Figure 7.** Comparison between area growth rate over time. **Source:** SEGETH 1960-2015(area growth).

**Table 6.** Population growth rate over time in Brasília.

| Year | Population | Urban Area (km$^2$) | Area Growth Rate (%) | Density (Pop./km$^2$) | Pop. Growth Rate (%) |
|------|------------|---------------------|----------------------|-----------------------|----------------------|
| 1960 | 139,796 | 22.78 | - | 6136 | - |
| 1964 | 335,461 | 56.99 | 1.50 | 5886 | 1.39 |
| 1975 | 904,814 | 181.86 | 2.19 | 4975 | 1.69 |
| 1982 | 1,364,000 | 240.11 | 0.32 | 5680 | 0.50 |
| 1991 | 1,601,094 | 291.61 | 0.21 | 5490 | 0.17 |
| 1997 | 1,821,946 | 393.07 | 0.35 | 4635 | 0.13 |
| 2009 | 2,606,885 | 537.41 | 0.37 | 4850 | 0.43 |
| 2015 | 2,906,574 | 569.59 | 0.11 | 5102 | 0.11 |

**Source:** IBGE National Census 1960-2015.

The new plans insisted on low/medium density and predominant residential projects on the east side across the lake—an area where an increasing number of condominiums has been observed in recent years. At the same time, the zoning plan was more restrictive, strictly prescribing areas of environmental concern, mostly located on the east side, aiming toward the preservation of agricultural land.

Regarding transportation strategy plans, the government launched the Urban Transportation Plan (law no. 4.566/2011), establishing the plan for "Integrated Brasília," in which the entire transport network of the territory was to be combined into a single regional system, integrating the itineraries of both the bus and subway systems. The plan established "Travel Generator Poles," intending to optimize the impact of mass transport on surrounding urban circulation rather than focusing on the formation or consolidation of these areas [61]. In terms of the area covered in the new plan, the project established both the territory of the Federal District and the eight municipalities in the surrounding state of Goiás as the target area of influence.

According to the last household travel survey conducted in Brasília (2000 and 2009), the implementation of a subway system and improvements on public transport do not show significant changes on transport mode choice within the Federal District. The percentage of inhabitants using public transport increased from 36.6% in 2000 to 41.0% in 2009, while the use of private automobile practically remained the same in 9 years, from 50.9% to 51.0% [74]. This inexpressive change in the transport mode choice could be partially explained by the distribution of land use in the region (see Figure 8). The signs of functionalism are still present in Brasília, where the separated land use patterns do not contribute to reducing vehicle travel and increase the use of alternative modes, particularly walking.

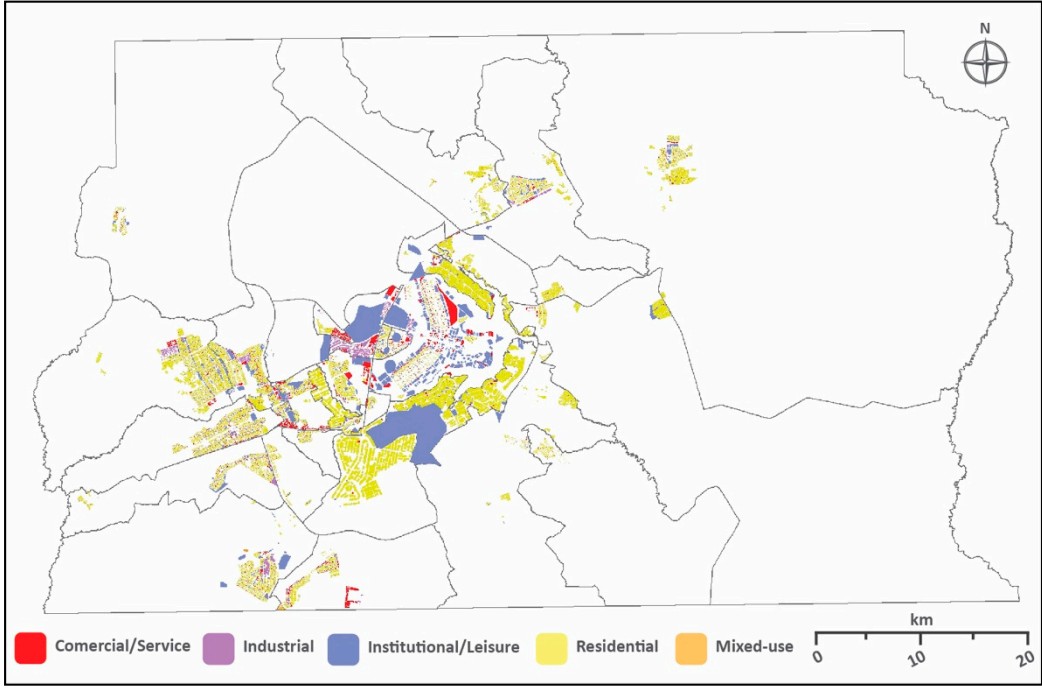

**Figure 8.** Built area land use in the Federal District (2017). **Source:** Adapted from Urban Land and Territorial Tax Report (SEGETH, 2017).

## 5. Conclusions

### 5.1. Summary and Conclusion

This study examined the evolution of urban spatial structure in Brasília focusing on the role of urban development policies. From the pilot plan of Lucio Costa (1957) to the most recent land use

plan of 2009, this research examined the evolution of urban spatial structure from a modernist and centralized policy to a strategic and decentralized policy in metropolitan Brasília area. The evolution of the urban form of the Federal District reflects a changing posture due to public policies.

Based on the comparison and analysis of historical facts, collected data, and various theories, political realities from the earliest stage of the development of Brasília must be considered as the foundation of its current spatial structure. The urban expansion of Brasília into peripheral areas occurred before the end of construction of the pilot plan due to public policies that aimed to preserve the original design via the rigid control of land use within the territory. Rapid population growth, however, created additional pressure for residential space in both inner and peripheral areas of the region. Moreover, by not accommodating different social classes and enabling the fostering of social relations, Brasília emerged as an embodiment of top-down planning. In this arrangement, it is possible to observe the origins of social segregation and fragmentation in Brasília, both founded on the intrinsic functional hierarchy in the original plan of Lucio Costa. Accordingly, the social driving forces observed in this study influenced new urban development in both directions—toward decentralization and against decentralization—depending on which social groups or political parties were involved at specific periods in the history of the urban development of Brasília.

During its early years of establishment, the Federal District faced a period of confused territorial governance due to unstable political circumstances on both local and national scales. Brasília went through a long period without any systematic regional approach for development. In addition, local authorities needed to address the problem of illegal residents through the creation of satellite cities far from the central area. These practices resulted in a disconnected network of districts without a proper integrated infrastructure and caused unnecessary consumptions of land. These problems were mitigated only after the implementation of both the Special Program for the Geoeconomic Region of Brasília (1975) and the Territorial Organization Structural Plan (1978)—the first two signals of strategic development.

The most important aspect observed in this study is the impact of territorial governance on the evolution of urban form in Brasília. In Brazil, urban management tends to be conducted without integrated strategies or instruments of action. This sectoral approach, which accumulates various urban policies that focus on the same territorial base, leads to spatial fragmentation. This fragmentation undermines the unity of the territory. The unity of the territory, observed in the past 40 years, is essential for the local economy and is an important issue observed in prior purpose-built capital cities, especially in secondary capital cities such as Brasília.

## 5.2. Discussion and Policy Implications

Although we agree that the local government's autonomy is undoubtedly a form of democratic progress due to the new federal constitution of 1988, the Brasília metropolitan area should be coordinated by regional planning, including central governments and local authorities. Additionally, in order to achieve proper territorial governance, this study considers it essential to give more administrative autonomy to urban development departments in local governments. The current administrative structure with urban development strategies, which is interrelated to political interests, is unfavorable to long-term urban sustainable development.

Furthermore, it is important to reinforce the policy of mixed-use development. Although mixed-use practices have gained attention among the current practices of the master plan, they remain poorly implemented. Transportation infrastructure investments exclusively concentrated on public transportation, without integrating diverse sectors such as housing and land use. As a result, housing and land use would never result in achieving sociosustainable development. Moreover, following other purpose-built capital cities, such as Putrajaya, the segregation resulting from prior housing policies, observed through the demographic division between satellite cities and Brasília, must be avoided in future development plans in Brasília or any other new capital city under development, such as Sejong in South Korea, and the yet-unnamed new capital city of Egypt.

This study indicated the important role of public policies on the evolution of the metropolitan Brasília area. The next step would be the coordinated developments among local governments and the Federal District through long-term strategies to achieve sustainable growth. It must be done without reducing the power of the municipality. Instead, public policies should strengthen existing relationships between stakeholders, looking for better coordination of interests, and improving the adaptability of governance processes in relation to socioeconomic and environmental demands. This study contributes toward a better reading of long-term urban development policies and their effects on the spatial structure of Brasília over time. The comparison with similar purpose-built capital cities presents similarities that must be considered when new urban settlements are developed. The findings presented here, therefore, add to the recent literature arguing that strategic development and flexibilization are fundamental for the changing needs of new capital cities.

Due to some limitations in this study, a few questions remained unanswered. It was not possible to conduct an empirical analysis of the impact of spatial structure on housing, transportation, environment, quality of life, and so on. Thus, new studies should address the consequences of the evolution of metropolitan Brasília.

**Author Contributions:** C.C., the leading author, initially conceived and designed the research. He performed data analysis and original draft preparation under the supervision of the corresponding author. S.L., a corresponding author, developed the original idea of this study and provided suggestions for the overall analysis. All authors contributed to manuscript preparation and discussed the results.

**Funding:** This research received no external funding.

**Acknowledgments:** Parts of this work were presented at the 2017 Fall Congress of the Korea Planning Association (KPA) held at Daegu Hanny University, South Korea and at the 2018 Fall Congress of the Korea Planning Association (KPA) held at the National University of Transportation in South Korea.

**Conflicts of Interest:** The authors declare no conflict of interest.

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
