# Peer review of "The Evolution of Urban Spatial Structure in Brasília: Focusing on the Role of Urban Development Policies"

_sustainability, doi:10.3390/su11020553_

Reviewer 1 Report

The article is interesting and the topic is worthy of research. Nevertheless, I find important shortcomings both at the methodological level and at the level of review of the state of the art. In this sense the article has to be seriously improved from a scientific point of view to be considered for publication in Sustainability journal in my opinion.Therefore, I recommend a major revision prior to its consideration, given that a series of methodological concretions are necessary to verify the validity of the scientific approach. Next, I detail the main problems and insufficiencies detected in the article:

METHODOLOGY

The methodology chapter is quite poor and not very illustrative of the analysis that has been carried out. It only includes in fact the origin and the sources of the data used. No methodology for scientific analysis is explained, nor is it detailed on the basis of which numerical evaluation procedure the results of the study will be obtained. In this sense, there is hardly any connection between the methodology section and what seems to be the results section. This section must be seriously improved detailing the numerical analysis carried out to improve the scientific soundness of the manuscript.

RESULTS 

The article presents sometimes a confusing structure. Chapter four seems to raise the numerical results of the investigation. Nevertheless, it is quite confusing in its structure and does not clearly separate the results from an objective scientific analysis, from the considerations and subjective contributions made by the authors. This section should be made more clearly, transferring those issues that do not respond to the simple description of the results obtained to a discussion chapter.

INTRODUCTION & STATE OF THE ART REVIEW 

The article presents a confusing structure again. It starts with a brief general introduction and continues with a specific review of Brasilia case study, to then resume again a more in-depth review in a literature review chapter, which is strange. In this sense, I recommend merging chapters one and two, so that a general review of the issue is carried out in a first subsection of the new chapter with a thorough review of the state of the art with international references, and then present in a second subsection the case study of Brasilia with its specific references related to Brazilian studies. In addition, there are hardly any international references of recent research of similar cases. The authors indicate that there are some cases in which a city is created from scratch, however there is no reference in the whole text to make a certain comparison. I suggest including references to other case studies created from scratch to strengthen the review of the art and expand the international interest of the article (e.g. García-Ayllón, S. La Manga case study: Consequences from short-term urban planning in a tourism mass destiny of the Spanish Mediterranean coast. Cities 2015, 43, 141–151).

DISCUSSION 

The authors merge the chapter of conclusions and discussion in a single chapter. However this section basically contains only a summary as a conclusion of the article. Therefore, there is no clear debate about the implications of the results obtained in the investigation. I recommend generating a specific chapter of discussions in which the implications of the results obtained are discussed. This chapter has to address the extent to which the work done has contributed to improving knowledge of the issues related to the urbanization process in Brasilia (of which there are numerous previous works in the scientific literature) and for example which are (at a more general level) the problems related to sustainability over the creation from scratch of a city with the conditioning factors of Brasilia to justify its interest for Sustainability readers.

Author Response

please see the attached response letter.

Reviewer 2 Report

This paper's stated aims are: to analyse the interrelation between urban planning and spatial structure over time; and to understand the role of urban development policies on the spatial organisation of Brasilia, the capital of Brazil.

The literature review offers some useful background knowledge about the topic, but there is too much of paraphrasing or citing other’s work. A good review should be a synthesis of existing literature. It should identify research themes, research gaps and justify the approach of the current study.

The research gap identified the authors is apparently at Line 173: "Although there are many studies on the characteristics of specific urban spatial structures, the outcomes of urban development policies are rarely evaluated."

There are studies that looked at the outcome of urban development policies, for instance:

Lai, L.W.C., Baker, M., 2014. The final colonial regional plan that lingers on: Hong Kong’s Metroplan. Habitat International 41, 216–228. https://doi.org/10.1016/j.habitatint.2013.08.001

Also, there is a lot of attention to master planned capital cities, and evaluation of them, to name a few:

Gordon, D.L.A., 1998. A City Beautiful plan for Canada’s capital: Edward Bennett and the 1915 plan for Ottawa and Hull. Planning Perspectives 13, 275–300. https://doi.org/10.1080/026654398364455

Neutze, M., 1987. Planning and land tenure in Canberra after 60 years. Town Planning Review 58, 147. https://doi.org/10.3828/tpr.58.2.l3753214v5313472

 There are also studies that compared multiple master planned capitals

Gordon, D.L.A., 2002. Ottawa-Hull and Canberra: Implementation of capital city plans. Canadian Journal of Urban Research 11, 179–211.

The authors should consider the above in the literature review as well. Currently, it is rather focused on Brazilian studies. A study in Phoenix (Luck and Wu, 2002) is cited, but that is a detailed land use change analysis, which the authors did not carry out with the same detail.

The paper explored how planning ideas/movements influenced Brasilia’s planning and development and outlined its urban evolution over time. It is more akin to a historical review of how Brasilia evolved over time, added with spatial analysis (more about the location of urban growth, less about transport and land use changes) and offered some statistics (mostly population and urban area changes). These might be useful if framed as a Brazilian urban history study, but it is less useful to offer knowledge for urban scholars or practitioners outside Brazil. There is way too much emphasis on historical events, population/urban area changes, but too little about “What does it mean for urban planning (and sustainability?)” and “How can we do better?” Nor there is a clear conceptualisation or theorisation of purpose-built cities. Also, there is a lack of discussion about how purpose-built cities are different from organic ones? These discussions are vaguely covered in the conclusion, but it is rather shallow and does not illuminate the issue. External political events, on the other hand, is being portrayed as the dominant factor that has greatly affected the city’s evolution. If so does it mean spatial plans are not effective? Also with the normalisation of urban planning, is Brasilia doing it better now? And how about the future?

Also in Line 187 “Many cities evolve over time, but some are designed from scratch,” is not wrong, but a rather weak statement. Cities certainly evolve over time, but some are more purposely built, some are more organic. Currently, this paper focus on Brazil and Brasilia. It might be useful to offer comparison with (or at least draw parallels from) other similar purpose-built capitals, to name some: Washington (USA, 1800), New Delhi (India, 1912), Canberra (Australia, 1927), Islamabad (Pakistan, 1960), Putrajaya (Malaysia, 2002), and many more. This improves the contribution of this paper - to offer knowledge for future capital plans (or proposals) such as Sejong in Korea, Ciudad de la Paz in Equatorial Guinea, Egypt and Indonesia.

Further, there is no mentioning of the evolution of land use and transport changes in the paper. This is important. As the authors have pointed out - Brasilia was originally planned as a highway-based, car-oriented city, but later to accommodate urban growth, a metro is built (not explicitly mentioned in the paper, but clearly shown in Figure 7). Would this help make the city more balanced, accessible, and less car dependent? How about active travel opportunities? Is the city walkable/cyclable overall? The conclusion also mentioned mixed-use, there is however little analysis of land use mix in the analysis (only mentioned it was prohibited by the early architect, Line 276). The maps also do not show the degree of land use mixing.

Another thing is, the stated intention of Brasilia is to promote urban growth of central Brazil, is it successful? This might be slightly off-topic, but it is an important raison-d'etre of purpose-built capitals.

This paper has the potential to be more groundbreaking and with better contribution that helps to inform the development of future purpose-built capitals considering the case of Brasilia (and maybe drawing parallels with other international examples). This could be done by studying the Brasilia case and evaluate whether it's urban planning idea, concept, plans are effective/desirable with the lens of sustainability (including economic. social and environmental consideration. Sustainability is actually the main objective of this journal, but this has not been adequately referred to in the paper.

The writing should also be more concise, so as to provide room for the above-mentioned aspects that are not included currently.

 Minor comments:

 Figure 3 – It could be easier to understand the map by colouring the map by the four sectors

 Figure 4 – It might be better to compare first generation and second generation satellite cities, also, I do not understand why the blocks of built areas are related to paper Line 339-340 stating first generation cities are dormitory cities.

 Table 3 - The authors should clarify what unit is used for the monthly income (minimum wage).

 Table 4 – What time did these measurements take place? 2015? If so, please label it clearly on the title.

 Figure 5 – It is easier to understand if the authors could colour code the urban centres (circles in different sizes) into first and second generation as outlined in Table 4 and 5.

 Figure 6 – Please indicate the years of the four maps.

 Figure 7 – The map seems to be overlaid above the table.

 Line 360 – What does “superior” here mean? In relation to what other concepts that are considered “inferior”?

 Line 362 and Line 417: “Great external conditions” is a term coined by the authors. It could be a translation issue. Should it be called “Major external event?” It is up to the authors to decide what is the best. (Also relates to Figure 7).

Author Response

please see the attached response letter.

Round  2

Reviewer 1 Report

I observe a quite important effort from the authors to include all the reviewer´s suggestions. Although in the current context I still have some concerns about:

- the review of the state of the art: there still no appearing any reference to research studies of urban settlements created in XX century from scratch in unexpected environments, the authors only comment very superficially the case of Canberra and Washington (which has little connection with Brasilia case study since it is a city created 300 years ago with a traditional process as much in the world), but no references about other studies and methodological approaches (e.g. GIS) such as the one recommended by the reviewer of cities/urban settlements issues created from scratch are cited.

- the methodology section: I still no having the linkage between methodology and the results section since the methodology one is only described in general way, and no traditional method of numerical analysis and assess is implemented. It is true that the authors use a relevant theoretical framework such as Hersperger et al., but in my opinion I think that a greater concretion in this field is necessary to make the reader easier to understand the process by which the results are arrived at and to highlight the contribution of the authors to the scientific field from a methodological point of view.

Reviewer 2 Report

Thank you for the authors' diligent effort in addressing the reviewers’ comments. The paper is now much improved.

However, there are some areas that the authors could further improve the paper. Some of the previous comments remain unaddressed.

The literature review remains disjointed as there is still a lot of paraphrasing. There should also be a more focused discussion and synthesis about the “field” of purpose-built cities.

A paper just hot-of-the-press could be handy for the authors:

Kaufmann, D. (2018). Capital Cities in Interurban Competition: Local Autonomy, Urban Governance, and Locational Policy Making. Urban Affairs Review. https://doi.org/10.1177/1078087418809939 (their literature review is concise and aims to the point!)

This paper explored locational policy agendas of Bern, The Hague, Ottawa and Washington DC (which are “secondary capitals” that are politically important, but are not “primary economic centres” of their countries (Zurich, Amsterdam, Toronto, New York). Brasilia is in the same boat I would think.

There is also a recent book dedicated to strategic planning for “purpose-built” cities - Siegel F.R. (2019) Strategic Planning for Urban Centers: Present and Future. In: Cities and Mega-Cities. Springer Briefs in Geography. Springer, Cham

For Table 3 – Thank you for clarifying it is in minimum wage in Reais. However, it looks like there are actually in  “1,000 Reais”. I don’t think “10 Reais” is the highest income bracket!

Figure 4 looks squashed. Maybe this is to do with the track changes.

Figure 6: Please add a km scale for the density map of Brasilia. For greater clarity, please also consider separating the map and the area/population graph.

The discussion/conclusion section(s) is much improved, but still lacks substance. It is also unusual to have “Summary and conclusion” (Section 5.1) before the “Discussion of Policy Implications” (Section 5.2). A conclusion should be a neat concise section that concludes the paper at the very end, instead of adding new discussion points.  How about calling Section 5 – Discussion and/or Concluding Remarks, 5.1 Discussion, 5.2 Limitations and Future Research? 

The authors could make the earlier sections (1 to 4) more concise and provide more synthesis instead of reporting other’s literature and the result. The freed up words/space can be used to make a more impactful discussion and conclusion. A discussion should also link to the literature – what are the paper's implications to prior research/findings? It should also set the scene for future research. I can see the authors are trying to do this, but it is not strong enough yet. Consider how the finding of this research can help other researchers. Ideally, a statement should also be made to highlight the contribution of this research – Is it conceptual? theoretical? methodological? or applied (such as for offering guidance to planners/policy makers?).

I would also think interviews with stakeholders who were/are involved with Brasilia development can be a helpful addition in future research (as seen in Kaufmann (2018) above). It could also be interesting to compare Brasilia with other purpose-built capitals/cities. There are some research methods used in urban/transport studies – for example policy or discourse analysis as in Leung A, Matthew I, Burke A Cui J (2017) The peak oil and oil vulnerability discourse in urban transport policy: a comparative discourse analysis of Hong Kong and Brisbane. Transport Policy.  https://doi-org/10.1016/j.tranpol.2017.03.023

Finally, the flow of writing and choice of words seem unnatural at times. I suggest the authors proofread the work more thoroughly. Or perhaps consult a native English speaker to help with editing.

If these comments are addressed, I think this paper would be of keen interest to urban scholar community and policy makers (especially those who are contemplating new purpose-built cities).

Minor comments/suggestions

Line 30: “Diverse countries” seems odd here. Perhaps you mean “a certain”? or “some”? (According to this Conversation piece https://theconversation.com/egypt-is-building-a-new-capital-city-from-scratch-heres-how-to-avoid-inequality-and-segregation-103402, they claim there are about 30 purpose-built national/sub-national capitals in the World, this article also used Brasilia's case for Egypt's new capital plan - to consider inequity issues).

Line 36:  “imperative” instead of “fundamental” ?

Line 61: Madaleno, I.M. Brasília: the frontier capital. Cities 1996, Volume 13:4, pp. 273-280. - this reference is rather old. Consider more recent ones, or simply summarise the trend of west-central region’s population percentage (before and after Brasilia’s founding) using census statistics.

Line 173: “terrible” is a hyperbole and subjective. Consider better wording (e.g. undesirable? negative?)

Line 243: “can be considered” instead of “are considered” is more appropriate. Government policies are of less influence in Anglo-Saxon (UK, USA, Australia, Canada, New Zealand, etc.) styles of planning, where private property rights are paramount.

Line 531: I don’t feel Brasilia (as a whole) became a “more compact urban centre” as it did not really fully contain urban growth to its CBD as seen in the diagrams. Or it is the entire Brasilia is more developed as a whole and become functionally polycentric?

Author Response

please see the attached file.

Round  3

Reviewer 2 Report

Thank you for the authors effort on this another round of review. The paper is now much improved and I recommend for publication after minor corrections:

I noted the following outstanding minor issues:

Line 94: According to Vale (21) – [ ] brackets should be used for consistency

Table 3: Thanks for the clarification and correction of the currency changes. The purpose of this table is to demonstrate the distribution of incomes of the satellite cities of Brasilia. Also considering the many changes in the currency and hyperinflation, so it is appropriate to use minimum wages are as the unit for income. Perhaps just to make it clearer, simply state 1 unit of minimum wage = Cr$4149,6 in the title or at the notes in the *.

Just noted, if 1 unit is Cr$4149,6 (which I confirm is correct from the decree too), should ¼ be Cr$1037,4? And 2 be Cr$8299,2?  I leave this to the authors to decide as I am not familiar with the situation in Brazil.

Figure 5: The footnote should be “Metro line works commenced in 1992 and completed in 2008”? (as you include the metro line for 1990-1994)

Figure 6: When comparing the density map here with Figure 7 (built area land uses), the large geographical units (administration regions/regiões administrativas) could be misleading due to modifiable areal unit problem. Consider using smaller census tracts for showing population density, or customised grids, if such data is available. For example, Figure 7 in the paper by Holanda et al. (2015) mapped density with smaller units and it looks quite different. The paper also feature some good visualisation of accessibility level and urban centres of Brasilia (and Sao Paulo).

Holanda et al. (2015) can be accessed at http://fredericodeholanda.com.br/textos/holanda_et_al_2015_brasilia_fragmented%20metropolis.pdf

Figure 7: The land uses of non-built area are not shown in this map. I think this map should be called “built area land use” (or something similar) instead of (general) “land use”?. Also, should the north sign be pointing 0 degree upwards here?

The authors should also proofread and clear up any remaining typos or mapping errors not detected by the reviewers.
